# Implications of *OPRM1* and *CYP2B6* variants on treatment outcomes in methadone-maintained patients in Ontario: Exploring sex differences

**Caroul Chawar**[1,2], **Alannah Hillmer**[1,2], **Amel Lamri**[3,4], **Flavio Kapczinski**[2], **Lehana Thabane**[4,5,6], **Guillaume Pare**[4,5], **Zainab Samaan**[2]*

1 Neuroscience Graduate Program, McMaster University, Hamilton, ON, Canada, 2 Department of Psychiatry and Behavioural Neurosciences, St. Joseph's Healthcare Hamilton, Hamilton, ON, Canada, 3 Department of Medicine, McMaster University, Hamilton, ON, Canada, 4 Population Health Research Institute, Hamilton, ON, Canada, 5 Department of Health Research Method, Evidence, and Impact, McMaster University, Hamilton, ON, Canada, 6 Father Sean O'Sullivan Research Centre, St. Joseph's Healthcare Hamilton, Hamilton, ON, Canada

* samaanz@mcmaster.ca

**Data Availability Statement:** Data used in this research cannot be shared publicly as they contain potentially identifying and sensitive participant

## Abstract

Genetic variants in the *OPRM1* and *CYP2B6* genes, respectively coding for an opioid receptor and methadone metabolizers, have been linked to negative treatment outcomes in patients undergoing methadone maintenance treatment, with little consensus on their effect. This study aims to test the associations between pre-selected SNPs of *OPRM1* and *CYP2B6* and outcomes of continued opioid use, relapse, and methadone dose. It also aims to observe differences in associations within the sexes. 1,172 participants treated with methadone ($n_{Male}$ = 666, $n_{Female}$ = 506) were included in this study. SNPs rs73568641 and rs7451325 from *OPRM1* and all the tested *CYP2B6* SNPs were detected to be in high linkage disequilibrium. Though no associations were found to be significant, noteworthy differences were observed in associations of *OPRM1* rs73568641 and *CYP2B6* rs3745274 with treatment outcomes between males and females. Further research is needed to determine if sex-specific differences are present.

## Introduction

### Background

Methadone maintenance treatment (MMT) targeted for patients with opioid use disorder (OUD) has been proven over time to decrease opioid cravings and use [1]. However, due to the chronic classification of OUD, MMT is not curative, but aims to maintain patients on a specific dose, controlling their opioid use and enabling them to regain stability [1–3]. Administered methadone binds to endogenous opioid receptors in the human brain, eliciting similar effects within the reward system as an opioid would, while suppressing withdrawal symptoms [4].

information. Please contact the co-chair of the Hamilton Integrated Research Ethics Board, Frederick Spencer, (https://hireb.ca/; email: fspence@mcmaster.ca) or the corresponding author (samaanz@mcmaster.ca) for data access information related to this manuscript.

**Funding:** ZS was partially supported for this work by CIHR (grant number PJT–156306; https://cihr-irsc.gc.ca/). The funders had no role in study design, data collection and analysis, decision to publish, or preparation of the manuscript.

**Competing interests:** The authors have declared that no competing interests exist.

Though effective in reducing opioid use, MMT has been observed to have interindividual variability in methadone's metabolism and methadone blood concentration for a given dose [5]. This can be potentially dangerous to patients, as prescribing physicians are unable to accurately predict the patient's reaction to a methadone dose prior to administering it. If the methadone dose administered is too low, the patient can be at a high risk of relapse [6, 7]. Alternatively, if the dose is too high, the patient might be at a risk of overdosing, if supplementing with other opioids [8]. As such, a genetic predisposition for individual-based MMT outcomes has been the focus of much research [9–12].

The opioid receptor proteins, encoded by the *mu opioid receptor 1* (*OPRM1*) gene, bind both endogenous and exogenous opioids, resulting in pain relief and feelings of euphoria [13]. Single nucleotide polymorphisms (SNPs) in *OPRM1* have been associated with the number of opioid receptors present and their ability to function [14]. *OPRM1* SNPs rs1799971 and rs1799972 have been previously implicated in opioid use disorder [15]. Interestingly, rs1799971, rs73568641, and rs10485058 have been associated with methadone plasma concentration, methadone dose, and opioid use changes [16].

The enzymes encoded by the *cytochrome P450 family 2 subfamily B member 6* (*CYP2B6)* gene are involved in metabolizing 2 to 10% of clinically administered drugs, including methadone [17]. SNPs in this gene can lead loss or gain of function of the encoded proteins, possibly resulting in altered drug metabolism [18]. Many *CYP2B6* SNPs have been implicated in altered methadone metabolism and plasma concentrations, most notably rs2279343 and rs10403955 [11, 19, 20]. Some studies have also found associations to adverse events in methadone patients, with rs8192719 and rs3745274 associated with overdose fatality [16, 21].

Disparities in opioid use patterns, health and social functioning, and polysubstance use in methadone patients have been observed between the sexes [22, 23]. Further, genetic differences between sexes have been detected in psychiatric disorders and traits, and studies have highlighted the presence of sex-dependent effects in models with common genetic variants [24, 25]. Though past studies have adjusted for sex in their analysis models, very few have been observed to assess the contribution of sex to the genetic predisposition to MMT outcomes using rigorous sex-based analyses, considering how findings might differ within males and within females.

Studying select *OPRM1* and *CYP2B6* SNPs in a European sample would allow us to not only confirm conclusions within the published literature but also test if the strength of these associations holds true to direct clinical MMT outcomes observable in patients, such as continued opioid use, relapse, and methadone dose. Additionally, having comparable male to female ratios within our sample enables us to robustly examine sex-based differences that have not been adequately highlighted in past studies.

## Objectives

This study aims to report new genetic associations that have not been tested previously, as well as analyze associations with biological relevance from previous literature within a larger sample of European descent. The objectives of this study are to:

1. Test the association between pre-selected *OPRM1* (rs73568641, rs7451325, rs10485058, rs1799971) and *CYP2B6* (rs2279343, rs10403955, rs8192719, rs3745274) SNPs and continued opioid use, relapse, and methadone dose in MMT patients; and

2. Determine if there are differences in associations within and between the sexes through sex stratification and exploratory SNP x Sex interaction analyses.

## Methods

This candidate gene study is reported according to Strengthening the Reporting of Genetic Association studies guideline, an extension of Strengthening the Reporting of Observational studies in Epidemiology statement [26]. An accompanying guideline checklist could be found in S1 File.

### Study design and setting

This research reports data collected by the Genetics of Opioid Addiction (GENOA) study, which is an observational cohort study of 1,536 participants recruited from Canadian Addiction Treatment Centres across Ontario, Canada [27]. Data collected at the baseline (enrollment in the study) are the primary sources of information used. The data used include socio-demographic, opioid use-related, and treatment-related information, as well as information obtained from urine toxicology screen (UTS) results and blood samples. UTS results were also collected 3 months prior to study enrollment and up to a 12-month follow up period for measuring treatment outcomes. UTSs testing for opioid use were conducted regularly, on a weekly/biweekly basis, with results reported at 3-month intervals for the GENOA study. The GENOA study was approved by the Hamilton Integrated Research Ethics Board (#11056). All the participants enrolled in the study provided written informed consent.

### Eligibility criteria

The participants selected for this study are those deemed eligible by the GENOA study eligibility criteria [27]. These required participants to be 18 years of age or older, have a Diagnostic and Statistical Manual of Mental Disorders [5th edition] OUD diagnosis, undergo an opioid substitution or antagonist therapy for OUD, and provide informed consent. Further inclusion criteria for all research questions addressed in this study include only participants who have provided a DNA sample and have received methadone as the primary opioid substitution or antagonist therapy.

For the measures of continued opioid use and relapse, participants must have had UTSs assessing the presence of opioids for a minimum duration of 3 months and 6 months, respectively. Participants taking prescription opioid medications were excluded due to the uncertainty of the opioid origin when reviewing the UTSs in these participants. These exclusion criteria did not apply to the methadone dose outcome measure as no UTSs were used for that set of analyses.

### Outcomes and quantitative variables

Outcomes measured in this study include the following:

1. Continued opioid use while on MMT, defined as any opioid positive UTS (including opiates and oxycodone) observed over a duration of 3 to 15 months. It was measured as a binary variable.

2. Relapse while on MMT, defined as an event of opioid positive UTS following at least 3 months of opioid negative UTSs. It was measured as a binary variable.

3. Methadone dose while on MMT, defined as the amount of methadone a patient is administered at the time of study recruitment in milligrams. It was measured as a continuous variable.

Covariates for the measures of continued opioid use and relapse that were accounted for in the statistical models included: sex, age in years, methadone dose in milligrams, duration on MMT in months, and 5 principal components accounting for differences in ethnicity. Covariates accounted for in the measure of methadone dose were sex, age, duration on MMT, weight in kilograms, and the principal components. For the sex stratified analyses, the same variables as above were included in the additive models.

Genetic variants tested were identified from literature reviews, systematic reviews, candidate gene studies and genome-wide association studies as those related to *OPRM1* or *CYP2B6* and associated with altered methadone metabolism, methadone plasma concentrations, methadone dose, opioid use, or other treatment outcomes. Details about each selected SNP are shown in Table 1.

## Data handling

DNA was extracted from blood samples and the genotyping was performed by the Genome Quebec Innovation Centre (Montreal, Canada) [30], using the Illumina Global Screening Array-24 v1.0. Standard genetic association study quality control checks were applied using PLINK v1.09 and the RStudio interface for R i386 3.5.1 [31–33]. Genotype imputation in participants of European ancestry (as confirmed by PCA, n = 1,226) was performed using the Haplotype Reference Consortium's data as a reference panel via Michigan Imputation Server, using EAGLE2 and Minimac4 [34–36]. Post-imputation variant filtering was conducted, excluding SNPs with Rsq quality metrics of less than 0.3 and/or minor allele frequencies lower than 0.05.

SNPs reported in high linkage disequilibrium ($r^2 > 0.2$) were pruned, keeping the SNP with the most reported clinical significance and published associations, as seen on NCBI's SNP database [37]. As such, *OPRM1* rs7451325, and *CYP2B6* rs2279343, rs10403955, and rs8192719 were excluded. HaploView was used to visualize SNPs in linkage disequilibrium and calculate r-squared coefficients [38].

A detailed description as well as a flowchart outlining the different steps conducted to reach the final sample size are available in S2 File.

## Bias

Measures were taken in this study to identify areas of bias and address them. However, there remained potential sources of bias that could not be avoided, and thus are reported here. Outcomes of continued opioid use and relapse were defined through UTSs as opposed to relying on patient self-reports to remain as objective and unbiased as possible. However, measures such as methadone dose and duration on MMT were self-reported, allowing for a potential of social desirability bias, where participants might provide false information in lieu of more accurate responses that might be viewed as less desirable. Social desirability bias could also have elicited differing responses within males and females as behaviours might seem more desirable in one sex but not the other [39]. In addition, the findings might be affected by volunteer bias, wherein the sample recruited could not have been representative of the entire OUD population receiving treatment. Furthermore, only participants of European ethnicities were included in the analyses conducted. This might result in data that are not generalizable or lack replicability in other ethnic populations. Lastly, since the nature of this study is observational, it is not possible to control for all variables present, and as such undetected biases could have contributed to the findings reported.

**Table 1. Selected SNP details and genotype counts in European participants from the GENOA study.**

| Gene | Chr: Position (GRCh37) | SNP ID | Genotypes** | Genotype count | MAF* | HWE p-value* | Previously associated trait | Reference |
|---|---|---|---|---|---|---|---|---|
| OPRM1 | 6:154025139 | rs73568641 | | | 0.152 | 0.151 | Daily methadone dose | [28] |
| | | | CC | 35 | | | | |
| | | | CT | 304 | | | | |
| | | | TT | 887 | | | | |
| OPRM1 | 6:154016517 | rs7451325 | | | 0.152 | 0.149 | Daily methadone dose | [28] |
| | | | CC | 35 | | | | |
| | | | CT | 303 | | | | |
| | | | TT | 888 | | | | |
| OPRM1 | 6:154445215 | rs10485058 | | | 0.132 | 0.901 | Opioid positive urine screens of methadone patients | [16] |
| | | | GG | 20 | | | | |
| | | | GA | 283 | | | | |
| | | | AA | 923 | | | | |
| OPRM1 | 6:154360797 | rs1799971 | | | 0.11 | 0.662 | Opioid use disorder | [29] |
| | | | GG | 16 | | | | |
| | | | GA | 237 | | | | |
| | | | AA | 973 | | | | |
| CYP2B6 | 19:41515263 | rs2279343 | | | 0.246 | 0.282 | Higher S-Methadone plasma concentrations | [19] |
| | | | GG | 67 | | | | |
| | | | GA | 470 | | | | |
| | | | AA | 689 | | | | |
| CYP2B6 | 19:41509438 | rs10403955 | | | 0.259 | 0.941 | Higher S-Methadone plasma concentrations; lower apparent clearance of S-Methadone | [19] |
| | | | GG | 83 | | | | |
| | | | GT | 470 | | | | |
| | | | TT | 673 | | | | |
| CYP2B6 | 19:41518773 | rs8192719 | | | 0.24 | 0.433 | Increased frequency in methadone fatalities | [21] |
| | | | TT | 65 | | | | |
| | | | TC | 458 | | | | |
| | | | CC | 703 | | | | |
| CYP2B6 | 19:41512841 | rs3745274 | | | 0.234 | 0.577 | Increased frequency in methadone fatalities | [21] |
| | | | TT | 63 | | | | |
| | | | TG | 447 | | | | |
| | | | GG | 716 | | | | |

MAF = minor allele frequency. HWE = Hardy-Weinberg equilibrium.

*Data from the GENOA study (N = 1226).

**Alleles are on the + strand.

## Statistical methods

Descriptive statistical analyses were conducted on the total samples and stratified by sex to describe the demographic and clinical characteristics of the sample. Continuous variables were expressed as means with standard deviations, while categorical variables were expressed as counts. Chi square tests were conducted for categorical variables and t-tests for continuous variables to measure differences between the sexes.

Separate regression analyses were performed to test the association between each set of gene SNPs and the outcomes of continued opioid use, relapse, and methadone dose. An additive genetic model was used for all variants and all tests. Logistic regressions were conducted

to test the associations of continued opioid use and relapse, with the analyses testing for the association of having the minor allele and the outcomes as specified earlier. A linear regression model was used to test the association of having the minor allele and the outcome of methadone dose. All covariates were adjusted for, measuring their associations with the outcomes of interest. Furthermore, identical but separate regression analyses were conducted for male and female subsets, respectively. For analyzing sex differences, interaction analyses were performed with SNP x Sex as the interaction term in the regression models.

Samples with missing outcome values were excluded from the analysis. For the logistic regression analyses, missing values for the covariates of methadone dose and duration on MMT were imputed via mean substitution, from the averages of the values calculated per analysis. The same method was used to impute for missing weight and duration on MMT values for the linear regression.

Bonferroni corrected p-values of P<0.017 for OPRM1 SNPs and P<0.05 for CYP2B6 SNPs were used as thresholds for significance. All statistical analyses were performed on PLINK v1.09 and the RStudio interface for R i386 3.5.1 [31, 32].

## Results

### Participants

Samples from 1,226 participants and 5,563,682 SNPs passed quality control checks and filtering after imputation. After sample data cleanup and applying eligibility criteria for each outcome of interest, 1,129 samples were analyzed for continued opioid use, 944 samples for relapse, and 1,165 samples for methadone dose (S2 File).

### Descriptive data

Participant demographics and clinical characteristics can be seen in Table 2. Of the 1,226 ethnically European participants, 57% were male and 43% were female. The majority of participants were never married, unemployed, on methadone, and not prescribed opioid medications. The mean duration on MMT, age of first opioid use, and total number of positive opioid urine screens, as well as continued opioid use and relapse outcomes, did not differ significantly between the sexes. The weight and dose of methadone administered were lower in females than males, as would have been expected, as individuals of lower weight tend to be prescribed lower doses of MMT. In addition, the ratio of employed to unemployed males (0.70) was significantly higher than that of females (0.37).

### Main results

Results of the sex-stratified association analyses between the *OPRM1* SNPs (rs73568641, rs1799971, rs10485058) and continued opioid use, relapse, and methadone dose are shown in Table 3. No associations reached the Bonferroni adjusted significance threshold of P<0.017. However, some near-significant associations were observed within females but not within males, notably regarding rs73568641. Allele C expressed a potential of decreased odds of continued opioid use within females [OR = 0.71, 95%CI = 0.47,1.07, P = 0.098]. Its presence also signified a potentially more pronounced decrease in methadone dose in females [β = -7.99, SE = 3.73, P = 0.033] than in males [β = -2.36, SE = 3.33, P = 0.48].

Results of the sex-stratified association analyses between the *CYP2B6* SNP rs3745274 and continued opioid use, relapse, and methadone dose are shown in Table 4. No associations were found to be significant (P<0.05). Nonetheless, a near-significant association between the

**Table 2. Characteristics of participants of European ancestry with available genotype data in GENOA.**

| | Total | Male | Female | p-value |
|---|---|---|---|---|
| **N (%)** | 1226 | 699 (57) | 527 (43) | |
| **Age in years[a], Mean (SD)** | 39 (11) | 40 (11) | 38 (11) | 9.25E-03* |
| **Weight in kg[b], Mean (SD)** | 80 (21) | 86 (20) | 72 (19) | 2.2E-16* |
| **Marital status[c], N (%)** | | | | 3.51E-03* |
| Common law | 236 (19) | 118 (17) | 118 (22) | |
| Divorced | 125 (10) | 77 (11) | 48 (9) | |
| Currently married | 144 (12) | 95 (14) | 48 (9) | |
| Never married | 555 (45) | 328 (47) | 227 (43) | |
| Separated | 134 (11) | 64 (9) | 70 (13) | |
| Widowed | 31 (3) | 15 (2) | 16 (3) | |
| **Employment[d], N (%)** | | | | 4.86E-07* |
| Employed | 430 (35) | 287 (41) | 143 (27) | |
| Unemployed | 793 (65) | 411 (59) | 382 (73) | |
| **Methadone dose in mg[e], Mean (SD), [Range]** | 75 (45), [1–400] | 78 (47), [2–400] | 71 (43), [1–280] | 6.28E-03* |
| **MAT[f], N (%)** | | | | 0.69 |
| Methadone | 1172 (96) | 666 (96) | 506 (96) | |
| Suboxone | 52 (4) | 31 (4) | 21 (4) | |
| **Duration on MMT in months[g], Mean (SD)** | 45 (48) | 45 (48) | 44 (49) | 0.74 |
| **Age of first opioid use[h], Mean (SD)** | 25 (9) | 25 (9) | 25 (9) | 0.93 |
| **Participant taking opioid prescription[i], N (%)** | | | | 0.83 |
| Prescribed opioids | 34 (3) | 20 (3) | 14 (3) | |
| Not prescribed opioids | 1192 (97) | 679 (97) | 513 (97) | |
| **Total number of opioid screens[j†], Mean (SD)** | 74 (35) | 74 (34) | 75 (35) | 0.57 |
| **Total number of positive opioid screens[k], Mean (SD)** | 13 (21) | 13 (20) | 13 (22) | 0.70 |
| **Continued opioid use outcome[l], N (%)** | | | | 0.32 |
| Continued opioid use | 893 (79) | 513 (80) | 380 (78) | |
| No continued opioid use | 236 (21) | 127 (20) | 109 (22) | |
| **Relapse outcome[m], N (%)** | | | | 0.30 |
| Relapse | 433 (46) | 251 (47) | 182 (44) | |
| No relapse | 511 (54) | 279 (53) | 232 (56) | |

[†]260 of reported total included participants screened only for opiates.

*Significant difference between males and females.

All means were calculated excluding missing values.

[a]Data available for $n_{Total}$ = 1226, $n_{Male}$ = 699, $n_{Female}$ = 527.

[b]Data available for $n_{Total}$ = 1216, $n_{Male}$ = 693, $n_{Female}$ = 523.

[c]Data available for $n_{Total}$ = 1224, $n_{Male}$ = 697, $n_{Female}$ = 527.

[d]Data available for $n_{Total}$ = 1223, $n_{Male}$ = 698, $n_{Female}$ = 525.

[e]Data available for $n_{Total}$ = 1166, $n_{Male}$ = 664, $n_{Female}$ = 502.

[f]Data available for $n_{Total}$ = 1224, $n_{Male}$ = 697, $n_{Female}$ = 527.

[g]Data available for $n_{Total}$ = 1162, $n_{Male}$ = 661, $n_{Female}$ = 501.

[h]Data available for $n_{Total}$ = 1197, $n_{Male}$ = 685, $n_{Female}$ = 512.

[i]Data available for $n_{Total}$ = 1226, $n_{Male}$ = 699, $n_{Female}$ = 527.

[j]Data available for $n_{Total}$ = 1226, $n_{Male}$ = 699, $n_{Female}$ = 527.

[k]Data available for $n_{Total}$ = 1218, $n_{Male}$ = 692, $n_{Female}$ = 526.

[l]Data available for $n_{Total}$ = 1129, $n_{Male}$ = 640, $n_{Female}$ = 489.

[m]Data available for $n_{Total}$ = 944, $n_{Male}$ = 530, $n_{Female}$ = 414.

**Table 3. OPRM1 SNPs and associated outcomes.**

| Outcome | SNP | N | Minor Allele | OR/BETA | 95% CI/SE | P |
|---|---|---|---|---|---|---|
| **Continued opioid use** | **rs73568641** | 1129 | C | 0.84 | 0.63, 1.10 | 0.21 |
| | *Male* | 640 | | 0.99 | 0.67, 1.45 | 0.95 |
| | *Female* | 489 | | 0.71 | 0.47, 1.07 | 0.098* |
| | **rs1799971** | 1129 | G | 0.97 | 0.70, 1.36 | 0.88 |
| | *Male* | 640 | | 1.11 | 0.72, 1.72 | 0.64 |
| | *Female* | 489 | | 0.87 | 0.51, 1.48 | 0.61 |
| | **rs10485058** | 1129 | G | 0.96 | 0.71, 1.30 | 0.78 |
| | *Male* | 640 | | 0.89 | 0.59, 1.36 | 0.60 |
| | *Female* | 489 | | 1.00 | 0.64, 1.57 | 0.99 |
| **Relapse** | **rs73568641** | 944 | C | 0.98 | 0.76, 1.25 | 0.85 |
| | *Male* | 530 | | 0.97 | 0.69, 1.34 | 0.82 |
| | *Female* | 414 | | 1.04 | 0.70, 1.54 | 0.86 |
| | **rs1799971** | 944 | G | 0.82 | 0.61, 1.90 | 0.17 |
| | *Male* | 530 | | 0.76 | 0.52, 1.09 | 0.14 |
| | *Female* | 414 | | 0.94 | 0.58, 1.52 | 0.80 |
| | **rs10485058** | 944 | G | 1.10 | 0.83, 1.44 | 0.51 |
| | *Male* | 530 | | 1.02 | 0.70, 1.49 | 0.91 |
| | *Female* | 414 | | 1.15 | 0.77, 1.73 | 0.50 |
| **Methadone dose** | **rs73568641** | 1165 | C | -4.24 | 2.49 | 0.089* |
| | *Male* | 664 | | -2.36 | 3.33 | 0.48 |
| | *Female* | 501 | | -7.99 | 3.73 | 0.033** |
| | **rs1799971** | 1165 | G | 0.20 | 2.90 | 0.95 |
| | *Male* | 664 | | 2.59 | 3.76 | 0.49 |
| | *Female* | 501 | | -4.92 | 4.63 | 0.29 |
| | **rs10485058** | 1165 | G | -0.45 | 2.72 | 0.87 |
| | *Male* | 664 | | -0.50 | 3.69 | 0.89 |
| | *Female* | 501 | | 0.24 | 4.00 | 0.95 |

The minor alleles are also the reference and tested alleles. OR is odds ratio and BETA is the beta coefficient for the regression. 95% CI is the 95% confidence interval levels (lower, upper) and SE is the standard error. All results reported are odds ratios and 95% confidence intervals, except for the methadone dose outcomes, which are BETA coefficients and standard errors. P is the p-value for the t-statistic. The significance threshold is $P < 0.017$.

*$P < 0.1$.

**$P < 0.05$.

T allele of rs3745274 and continued opioid use within males [OR = 0.73, 95%CI = 0.52, 1.014, P = 0.06] was observed.

Exploratory analyses showcasing differences in associations between males and females were conducted. No significant results are reported. For detailed results see Tables G and H in S2 File.

## Discussion

### Key results

This study did not observe any associations that reached the significance threshold set. However, differences in the levels of significance within males and females were detected. Females with the C allele of *OPRM1* rs73568641 showed higher significance levels and stronger protective properties towards continued opioid use than males, as well as a potentially decreased

**Table 4. CYP2B6 SNPs and associated outcomes.**

| Outcome | SNP | N | Minor Allele | OR/BETA | 95% CI/SE | P |
|---|---|---|---|---|---|---|
| **Continued opioid use** | **rs3745274** | 1129 | T | 0.82 | 0.64, 1.05 | 0.11 |
| | *Male* | 640 | | 0.73 | 0.52, 1.01 | 0.06* |
| | *Female* | 489 | | 0.95 | 0.66, 1.37 | 0.80 |
| **Relapse** | **rs3745274** | 944 | T | 0.91 | 0.73, 1.14 | 0.42 |
| | *Male* | 530 | | 0.86 | 0.64, 1.16 | 0.32 |
| | *Female* | 414 | | 1.07 | 0.76, 1.49 | 0.71 |
| **Methadone dose** | **rs3745274** | 1165 | T | 1.26 | 2.17 | 0.56 |
| | *Male* | 664 | | -1.17 | 2.99 | 0.70 |
| | *Female* | 501 | | 4.19 | 3.18 | 0.19 |

The minor alleles are also the reference and tested alleles. OR is odds ratio and BETA is the beta coefficient for the regression. 95% CI is the 95% confidence interval levels (lower, upper) and SE is the standard error. All results reported are odds ratios and 95% confidence intervals, except for the methadone dose outcomes, which are BETA coefficients and standard errors. P is the p-value for the t-statistic. The significance threshold is $P < 0.05$.

*$P < 0.1$.

methadone dose. However, the T allele of *CYP2B6* rs3745274 in males showed potential for being more protective and significant when it came to continued opioid use.

## Interpretation

The possible involvement of the C allele of *OPRM1's* rs73568641 in a decreased chance of opioid use and/or decreased methadone dose in females suggests the involvement of *OPRM1* gene in not only opioid use disorder, but also treatment outcomes. The similar direction of association observed with respect to continued opioid use and methadone dose is interesting given that previous research has reported that higher methadone doses are more effective at decreasing opioid use while on MMT [40]. However, since the variable of methadone dose was accounted for in the analysis model of continued opioid use, the results of the associations can be viewed as independent. When compared to the literature, these associations conflict with the only other published findings. *OPRM1* rs73568641 (allele C) seems to have an opposite effect in an African American population [28]. In a genome-wide association study subset (n = 383), it was found to slightly increase daily methadone dose [β = 0.681, P = 2.81E-08]. Unfortunately, no conclusions could be drawn due to the possibility that the differences observed between these findings could be a result of the ethnic contribution to the genetic makeup. This highlights the importance of ethnically diverse research and how interindividual differences of patients of different ethnic backgrounds could play a role in patient treatment outcomes.

While the role of the *CYP2B6* rs3745274 SNP was not determined in this study with regards to an MMT outcome, other studies have reported evidence of association across different haplotypes of *CYP2B6*, especially those where this SNP is found, and plasma methadone concentrations. In a pharmacokinetics study, *CYP2B6*6* carriers were observed to have higher S-methadone plasma concentrations than non-carriers [41]. It was also determined that *CYP2B6* inhibition reduces methadone clearance and increases plasma methadone concentrations [41]. This was further supported by other studies where *CYP2B6*6* was shown to have slower S-methadone intravenous clearance, slower R- and S-methadone oral clearance, higher plasma concentrations, and lower methadone dose requirements in carriers [20, 21, 42]. However, given methadone's racemic mixture and findings supporting R-methadone's heavier contribution to opioid effects, more evidence on the genetic effects on R-methadone metabolism is

required [5]. When comparing *CYP2B6* rs3745274 to literature findings on other treatment outcomes, the T allele seems to be associated with an increased frequency in methadone fatalities (P = 1.2E-03) in a sample of European ethnicity (n = 125) [21]. Though these fatality findings support the discussed literature, as a higher plasma concentration of methadone could also have negative effects and risks, such as death, the differing sample sizes of control and methadone-only groups (n = 255 and n = 125, respectively) could have contributed to such results.

This study was unique in stratifying analyses by sex and observing differential findings for each sex. The sex-based differences observed in the strengths of the associations could not be fully attributed to sample size, as seen in the strength of *OPRM1* rs73568641's associations in females despite having a smaller sample size than their male subset's counterpart. This could be indicative of larger biology-based differences within the sexes, which could have influenced the results. Examples could be the differing *CYP* enzyme activities between the sexes that could affect drug metabolism, or neuroanatomical differences in the dopaminergic pathway that can influence the effects of a drug on the system [43, 44]. It is also possible that gender construct and its implications can affect the results, even if indirectly. Women are more likely to become dependent on prescribed opioids than males, experience faster dependence progression rates, and have higher relapse rates [23, 24]. Men, on the other hand, report higher prevalence cannabis use and are more likely to be employed and financially secure [22, 45]. These are only a few examples of how the behavioural and social functioning implications associated with gender can influence phenotypes measures, such as continued opioid use and relapse.

## Limitations and generalizability

Aside from the sources of bias discussed earlier, some limitations in this study were faced and need to be addressed. Firstly, the findings are specific to a sample of European ethnic descent, making them not generalizable to samples of other ethnicities. Similarly, the sex-specific results may not be comparable to other study findings that do not conduct sex-stratified analyses. Another limitation is that there was a high degree of missingness within the data with respect to the measure of relapse, resulting in a smaller sample size for that set of analyses. Though a power analysis was conducted for the original GENOA project, it is not applicable due to the different SNPs analyzed in this specific study. Additionally, due to a lack of a reported and reliable effect size in the literature and the disputably misleading results of a post-hoc power analysis, an informative power calculation could not have been conducted [46]. Further data missingness was observed in the UTS results reported across the sample population. As the duration of UTS result collection ranged from 3 to 15 months, the outcomes of continued opioid use and relapse were not consistently measured. However, given that and the inevitable variability in how long participants had been on MMT, the duration on MMT was accounted for in all statistical models. An additional data-related limitation includes the inability to accurately use methadone dose as an indicator of treatment response in MMT patients. This is mostly due to the fact that patients on MMT could be at any of the induction, treatment, stabilization, or tapering stages, each of which characterized by a variable pattern of methadone dose administration. This participant variability also plays a role in the measurement of the relapse outcome, posing a challenge in accounting for all participants including those with some breakthrough opioid use while on treatment. Finally, since the exploratory between-sex analyses were insignificant, the interpretation of the sex-stratified results are made with caution. Though an insignificant interaction term could be interpreted as an absence of a difference between males and females, it could also be highly indicative of an under-powered study.

## Conclusion

Given that the study had a larger sample size than most similar published research within this field, it was able to address a gap in the genetics of MMT research. Though none of the results were significant, this study identified a need for ethnically diverse research, and uncovered the important contribution sex measures have towards outcomes of continued opioid use and methadone dose in MMT patients. Future recommendations towards more powered studies including sex in the analysis models are made.

## Supporting information

**S1 File. STREGA checklist.**
(DOCX)

**S2 File. Appendix.**
(DOCX)

## Acknowledgments

We would like to thank the GENOA research study team for their time and efforts in recruitment, as well as Canadian Addiction Treatment Centres and their patients for their collaboration and for making this study possible.

## Author Contributions

**Conceptualization:** Caroul Chawar, Zainab Samaan.

**Formal analysis:** Caroul Chawar, Alannah Hillmer.

**Funding acquisition:** Zainab Samaan.

**Investigation:** Zainab Samaan.

**Methodology:** Caroul Chawar, Alannah Hillmer, Amel Lamri, Flavio Kapczinski, Lehana Thabane, Guillaume Pare, Zainab Samaan.

**Project administration:** Caroul Chawar.

**Resources:** Zainab Samaan.

**Software:** Alannah Hillmer, Amel Lamri.

**Supervision:** Zainab Samaan.

**Visualization:** Zainab Samaan.

**Writing – original draft:** Caroul Chawar.

**Writing – review & editing:** Caroul Chawar, Alannah Hillmer, Amel Lamri, Flavio Kapczinski, Lehana Thabane, Guillaume Pare, Zainab Samaan.

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
