## [Decision Letter · Decision Letter 0]

14 May 2021

PONE-D-21-11145

Implications of *OPRM1* and *CYP2B6* variants on treatment outcomes in methadone-maintained patients in Ontario: Exploring sex differences

PLOS ONE

Dear Dr. Samaan,

Thank you for submitting your manuscript to PLOS ONE. After careful consideration, we feel that it has merit but does not fully meet PLOS ONE’s publication criteria as it currently stands. Therefore, we invite you to submit a revised version of the manuscript that addresses the points raised during the review process.

We look forward to receiving your revised manuscript.

Kind regards,

Huiping Zhang

Academic Editor

PLOS ONE

Journal Requirements:

2.We note that you have indicated that data from this study are available upon request. PLOS only allows data to be available upon request if there are legal or ethical restrictions on sharing data publicly. For information on unacceptable data access restrictions, please see http://journals.plos.org/plosone/s/data-availability#loc-unacceptable-data-access-restrictions.

Reviewers' comments:

Reviewer's Responses to Questions

**Comments to the Author**

1. Is the manuscript technically sound, and do the data support the conclusions?

Reviewer #1: Yes

Reviewer #2: Partly

Reviewer #3: Partly

2. Has the statistical analysis been performed appropriately and rigorously? 

Reviewer #1: Yes

Reviewer #2: I Don't Know

Reviewer #3: Yes

3. Have the authors made all data underlying the findings in their manuscript fully available?

Reviewer #1: Yes

Reviewer #2: Yes

Reviewer #3: No

4. Is the manuscript presented in an intelligible fashion and written in standard English?

Reviewer #1: Yes

Reviewer #2: Yes

Reviewer #3: Yes

5. Review Comments to the Author

Reviewer #1: The authors performed a genetic association study of methadone maintenance treatment (MMT) of opioid use disorder among ~1100 persons of European origin (~65% male) who had urine drug screen (UDS) data and methadone plasma levels available over a 3-15 month period, across a consortium of MMT clinics in Canada. They did array genotyping which yielded ~5 million SNPs after imputation. Despite this wealth of genetic data, they report on 3 OPRM1 and 2 CYP2B6 SNPs. There were 3 phenotypes: methadone plasma levels, any lapse to illicit opioid use (a single + UDS) and relapse (a single + UDS after at least 3 months of – UDS results). Logistic regression and linear regression analyses were implemented. Sex-specific results are reported. There were no statistically significant results. The following comments are relevant:

1. The authors must be clear about whether they analyzed (or plan to analyze) the methadone dose and other phenotypes using other SNPs, or whether they restricted analysis to only those 5 SNPs reported here. This has implications for multiple hypothesis testing.

2. The authors should consider analysis of relapse as a continuous variable, using % UDS +, instead of defining relapse as a single + UDS.

3. ~10-15% of the sample had missing data, but is it not possible to obtain the methadone dose or weight from the EHR?

4. The authors should report the numbers of patients in each of the categories for the two binary phenotypes in Table 2.

5. The authors might consider the use of generalized estimating equations in analysis of these data.

Reviewer #2: Comments to the Authors:

This manuscript employs a relatively large population of European MMT patients to examine the pharmacogenetic effects of variants in OPRM1 and CYP2B6 on MMT outcome measures. The authors further explore the effect of sex on these outcome measures (which the authors state has not been done in previous studies in the literature). While enthusiastic about the data set, the reviewer has reservations about the lack of information in the description of the study, the statistical methods and about the outcome measures used. Specifically, the outcome measures used by these authors are not consistent with those used in some previous literature, and, thus, cannot be used as a test of replication of previous findings (which is a stated objective of this manuscript). The reviewer believes that although the data analysis was likely done correctly, the manuscript could be improved before publication.

Major issues:

The authors state that sex-based analyses have not been done or have been overlooked by past studies. The reviewer finds the literature contains at least one study (Crist et al, 2018a) on OPRM1 genotypes and MMT efficacy to have used sex as a covariate in their statistical analyses. Although the Crist et al, 2018a study had fewer participants on MMT, sex was considered in their analysis, and the authors found no effect of sex on the measured outcome.

The authors state that the objective of the present study is to “replicate findings from the literature within a larger sample of European descent”. However, the defined outcome measures in the current study are not equivalent to those used in previous studies in the literature. The authors should reference those studies in the literature in which their outcome definitions have been used. For example, these authors employ binary data, rather than percent positive UTS over time by rs10485058 genotype (Crist et al, 2018a), as their outcome measure for “continued opioid use”. In addition, the authors use initial methadone dose at intake, rather than maximum methadone dose (Crist et al, 2018b; ref 28 - Smith et al, 2017), as their outcome measure (but, see question below about methadone dose at time of GENOA enrollment).

How long was each participant on MMT before enrollment in GENOA? The average time on MMT in the GENOA study was 3.7 years (Table 2), but the standard deviation is quite large. Did participants in the current study need to be on MMT for a minimum defined time to be included? If so, what was that minimum time requirement? The authors state that “primary sources of information used” were “data collected at baseline (enrollment in the study), 3 months prior to study enrollment, and up to 12 months follow up”. So, does the current manuscript encompass only data from a 15 months period that includes 3 months before enrollment into GENOA and 12 months of follow-up of participants after GENOA enrollment? Was UTS data available before enrollment of participants in the GENOA study, specifically at 3 months prior to enrollment? Were all participants enrolled in GENOA on a stable (maximal) dose of methadone at the time of enrollment, or were some participants enrolled before or during methadone titration up to a stable dose?

The authors should indicate how often UTS were done throughout the MMT period. Were they weekly or monthly? How were missed UTS appointments handled? Were they treated as missing or as positive? The definitions of “continued opioid use” and “relapse” do not seem rigorous enough because some level of continued opioid use is anticipated in real world settings of MMT, especially at the start of treatment. The reviewer suggests an analysis using percent positive UTS over time by genotype to determine MMT efficacy (see Crist et al, 2018a) especially at the start of MMT. Similarly, depending on the time on MMT of patients in the current study, the definition of relapse could be defined as more than 1 positive UTS after 3 months of clean UTS because some “breakthrough use” is to be expected as the clinician titrates the methadone dose to an effective level. The maximum dose of methadone is a better indication of therapeutic dose than the initial dose of methadone at intake (but, see question above about stable dose of methadone at time of GENOA enrollment). This is especially true for pharmacokinetic metabolic gene analysis, such as CYP2B6.

Regarding the discussion of SNP rs3745274 (*9 variant of CYP2B6), it is the R-enantiomer of methadone that is the therapeutic enantiomer and binds 50 times more strongly to the MOR (encoded by OPRM1), not the S-enantiomer. Because the authors did not determine the effect of rs3745274 on the R:S-enantiomer ratio, it is speculative to suggest that “[its] effect on continued opioid use in MMT patients could be explained as a decrease in the CYP2B6 gene activity, which could increase plasma methadone concentrations and subside the need for additional opioid intake.” The S-enantiomer is metabolized by CYP2B6, but the *9 variant is predicted to be benign by SIFT and PolyPhen, and was found not to significantly alter mean plasma methadone or the methadone:EDDP ratio (Figure 1 in reference 21 – Ahmad et al, 2017). Also, in Ahmad et al, the significant finding for rs3745274 is likely due to a skewing of the minor allele frequencies in different directions among the controls and methadone groups (overall predicted MAF=27%, controls=22%, methadone=31%) due to sampling error in small group sizes. (The reviewer also points out that the Ahmad et al study considered sex in their statistical analysis). Previous literature (ref 20 – Levran et al, 2013) indicates that it is the CYP2B6*6 haplotype (combined rs2279343 *4 with rs3745274 *9 genotypes) that led to differences in methadone dose requirements. The authors provide an image of the LD structure for CYP2B6 in their study population, but do not give the actual D’ and r2 values. From the image, it appears that although LD is high between *4 and *9 SNPs, it is not perfect. As such, to attempt to replicate previous findings in the literature, the authors should re-run the statistical analysis using the haplotype of these two SNPs, as was done in Levran et al, 2013.

What statistics program did the authors use to impute the missing covariate data? How many imputations did the authors run? If missing data imputations were run in R, the authors should state the package they used – for instance, did they use ‘norm’? Could the authors have instead used the 'keep-pheno-on-missing-cov' option for covariates in PLINK 1.07? Did the authors test models other than the ‘Genotypic’ (ADD) model? Other genetic models were used in previous literature (see Crist et al, 2018a). For covariates that were not significant in the full model, the authors should re-run the statistics without them in the model to determine whether the significance of the outcome measures change.

Minor issues:

Some of the references are incorrect. For example, reference 32 in the manuscript is supposed to reference R (page 11, line 208) but is listed in the references as Cahn et al, 2011, J. Wildl Manage 75(8):1753-66. Was this supposed to be reference 33 for R as listed in the references section?

The authors should state which analyzed SNPs were directly genotyped (on the chip) and which were imputed.

The authors should clarify what the reference condition was for each statistical analysis. For instance, was the reference condition defined as an individual ‘having the minor allele and being positive for the particular outcome measure’, or was it ‘having the minor allele and not being positive for the outcome measure’?

References mentioned (but not referenced in the manuscript):

Crist RC, Doyle GA, Nelson EC, Degenhardt L, Martin NG, Montgomery GW, Saxon AJ, Ling W, Berrettini WH. A polymorphism in the OPRM1 3'-untranslated region is associated with methadone efficacy in treating opioid dependence. Pharmacogenomics J. 2018a Jan;18(1):173-179. doi: 10.1038/tpj.2016.89.

Crist RC, Li J, Doyle GA, Gilbert A, Dechairo BM, Berrettini WH. Pharmacogenetic analysis of opioid dependence treatment dose and dropout rate. Am J Drug Alcohol Abuse. 2018b ;44(4):431-440. doi: 10.1080/00952990.2017.1420795.

Reviewer #3: This paper aims to test associations between several OPRM1 and CYP2B6 SNPs on several methadone maintenance treatment outcomes in males and females separately.

The strength of the study is the large sample size as well as the high number of females, which is rare in MMT studies. Contradictory findings have been published on these SNPs, mostly in smaller studies, thus a large replicate study is interesting.

My main concerns of the manuscript are as follows:

- In the abstract and in the results sections (also in the tables), the results are always described in such a way that they first seem to be statistically significant, while they are not.

- Most included patients were taking methadone but it seems from table 2 that a few took suboxone. Why they were not excluded as the study is on MMT?

- For the outcomes on continued opioid use and relapse, they should be clarified. The frequency of UTS should be indicated to better describe these outcomes. Is the duration of 3 to 15 months for all patients?

- For the methadone dose outcome, more information on how long the dose was unchanged before inclusion should be added. Detailed descriptive statistics of the dose should be added (range, normal distribution in the population, …).

- The outcome of the methadone dose is questionable as methadone metabolism displays a high interindividual variability and as patients might be in different phase of their MMT (treatment introduction, stabilization or slow tapering of the dose for example). Also, in certain MMT prescribing center, there is a maximal dose not be exceeded but that could remain insufficient for certain patients who therefore continue to use opiates.

In general, when discussing the individual SNPs, it will help to include the gene name before the rs number (ex. OPRM1 rs73568641)

In the study design and setting section (p.5, line 107): it is not clear which data was collected 3 months prior to study enrollment.

In Table 2, the significantly different variables between male and female should be indicated (*). The opioid prescription variable should be clarified.

In the discussion (p.16, line 258): no trend was observed for rs73568641 on relapse in female.

6. PLOS authors have the option to publish the peer review history of their article (what does this mean?). If published, this will include your full peer review and any attached files.

Reviewer #1: **Yes: **Wade Berrettini, MD, PhD

Reviewer #2: No

Reviewer #3: No

---

## [Author Response · Author response to Decision Letter 0]

10 Aug 2021

Reviewer #1: The authors performed a genetic association study of methadone maintenance treatment (MMT) of opioid use disorder among ~1100 persons of European origin (~65% male) who had urine drug screen (UDS) data and methadone plasma levels available over a 3-15 month period, across a consortium of MMT clinics in Canada. They did array genotyping which yielded ~5 million SNPs after imputation. Despite this wealth of genetic data, they report on 3 OPRM1 and 2 CYP2B6 SNPs. There were 3 phenotypes: methadone plasma levels, any lapse to illicit opioid use (a single + UDS) and relapse (a single + UDS after at least 3 months of – UDS results). Logistic regression and linear regression analyses were implemented. Sex-specific results are reported. There were no statistically significant results. The following comments are relevant:

1. The authors must be clear about whether they analyzed (or plan to analyze) the methadone dose and other phenotypes using other SNPs, or whether they restricted analysis to only those 5 SNPs reported here. This has implications for multiple hypothesis testing.

Response: 

Thank you for your review and feedback. We would like to clarify that we do not report on methadone plasma level, we report methadone dose and opioid use. This study is a hypothesis-driven study, the SNPs presented in this candidate gene study have been selected a priori to be tested for the continued opioid use, relapse, and methadone dose phenotypes. Analysis was restricted to the outlined OPRM1 SNPs (rs73568641, rs7451325, rs10485058, rs1799971) and CYP2B6 SNPs (rs2279343, rs10403955, rs8192719, rs3745274) as highlighted in the ‘Objectives’ section as well as in Table 1. Future analysis that may include hypothesis-free testing will not impact the current study, and any future publications will cite and refer to this study to ensure readers are aware of any relevant publications from the same study sample.

2. The authors should consider analysis of relapse as a continuous variable, using % UDS +, instead of defining relapse as a single + UDS.

Response:

Though the reviewer’s suggestion is insightful, the measure of relapse as a %UDS positive would not be an accurate representation of the event. The use of %UDS positive has been outlined in the previous literature as a way to measure continued opioid use. It would not be accurate in measuring relapse events as it is not indicative of an individual returning to opioid use (testing positive) following a period of no opioid use (testing negative). To best represent this pattern, relapse was determined a priori to be measured as any opioid positive UDS following at least 3 months of opioid negative UDSs. Having it as a binary variable as opposed to a continuous one allows for all incidences of relapse to be viewed as such, in a population that is not normally distributed. In addition, %UDS will be challenging to interpret as how much change is clinically relevant. 

3. ~10-15% of the sample had missing data, but is it not possible to obtain the methadone dose or weight from the EHR?

Response:

The methadone dose and weight measurements used in this study’s analyses were collected at the time of the participant interview as part of the greater GENOA study. The missing methadone dose data only contributed to less than 0.6% (7/1172) of the total analyzed population for the methadone dose phenotype. The missing weight data only contributed to 0.94% (11/1165) of the analyzed methadone dose sample, for which weight was considered a covariate. Since missingness in these variables was marginal and accounted for less than 10% of the sample population, any missing values were imputed using mean substitution when these variables were used as covariates, so as to not further affect the sample sizes.

For samples used in measures of continued opioid use (n=1129) and relapse (n=944), the smaller sample sizes were due to the exclusion of any participants who did not have UTSs assessing the presence of opioids for a minimum duration of 3 months and 6 months, respectively. Further exclusionary reasons included participants taking prescription opioid medications. These exclusionary reasons have been specified under the “Eligibility criteria” section of this manuscript and been discussed as limitations under “Limitations and generalizability”. 

4. The authors should report the numbers of patients in each of the categories for the two binary phenotypes in Table 2.

Response:

Thank you for the suggestion. We have revised Table 2 to also include a summary breakdown of patient outcomes for the continued opioid use and relapse phenotypes measured. 

5. The authors might consider the use of generalized estimating equations in analysis of these data.

Response:

The GEE model is best suited for longitudinal data and correlated observations; our data are binary and not looking at the individual in different time points for the primary study outcomes. Though analysis through generalized estimating equations might be helpful in determining the genetic variant-phenotype relationship in a longitudinal study such as the GENOA study, where outcomes of continued opioid use measured at multiple timepoints, the outcome is dichotomized, and it was not deemed the most appropriate method of analysis a priori. Generalized estimating equations are best used, for example, in family studies or multiple cohorts where multiple sources of correlation exist within the sample population. 

Reviewer #2: Comments to the Authors:

This manuscript employs a relatively large population of European MMT patients to examine the pharmacogenetic effects of variants in OPRM1 and CYP2B6 on MMT outcome measures. The authors further explore the effect of sex on these outcome measures (which the authors state has not been done in previous studies in the literature). While enthusiastic about the data set, the reviewer has reservations about the lack of information in the description of the study, the statistical methods and about the outcome measures used. Specifically, the outcome measures used by these authors are not consistent with those used in some previous literature, and, thus, cannot be used as a test of replication of previous findings (which is a stated objective of this manuscript). The reviewer believes that although the data analysis was likely done correctly, the manuscript could be improved before publication.

Response:

Thank you for your review and feedback. Please see below for detailed responses.

Major issues:

The authors state that sex-based analyses have not been done or have been overlooked by past studies. The reviewer finds the literature contains at least one study (Crist et al, 2018a) on OPRM1 genotypes and MMT efficacy to have used sex as a covariate in their statistical analyses. Although the Crist et al, 2018a study had fewer participants on MMT, sex was considered in their analysis, and the authors found no effect of sex on the measured outcome.

Response:

Though many studies adjust for sex in their analysis models, as was done in the referenced Crist et al study, the analysis is not designed a priori to test for differences within or between sex groups. To date and to the knowledge of the authors, very few have sought to specifically test the genetic contribution to MMT outcomes using sex-stratified analyses, considering findings separately within females and within males. The language used in the manuscript has been adjusted to reflect that. 

The authors state that the objective of the present study is to “replicate findings from the literature within a larger sample of European descent”. However, the defined outcome measures in the current study are not equivalent to those used in previous studies in the literature. The authors should reference those studies in the literature in which their outcome definitions have been used. For example, these authors employ binary data, rather than percent positive UTS over time by rs10485058 genotype (Crist et al, 2018a), as their outcome measure for “continued opioid use”. In addition, the authors use initial methadone dose at intake, rather than maximum methadone dose (Crist et al, 2018b; ref 28 - Smith et al, 2017), as their outcome measure (but, see question below about methadone dose at time of GENOA enrollment).

Response:

Though many studies have previously measured opioid positive urine tests as a continuous response outcome (i.e. the above referenced Crist et al, 2018a paper), our current study and previous literature have shown that opioid use is not normally distributed within this sample population, with a mean of about 20-50% of opioid urine screens testing positive (Bawor et al, 2015; Kamal et al, 2007; Hser et al, 2014). Please see a comment below elaborating on study data distribution and normality regarding positive UTS results. Numerous studies have also opted for measuring treatment response as a binary variable, as shown in Table 4 of Crist et al’s published review (Crist et al, 2018). 

With regards to the methadone dose measure, we have chosen to analyze this phenotype as the dose reported at enrollment since it was collected at the time of the participant interview as part of the greater GENOA study. We acknowledge that this measure of methadone dose might not be the most accurate indicator of MMT response as patients could be at different stages within their treatment. Though we have tried to account for this variability by adjusting for participant duration on MMT, we have discussed this limitation under the “Limitations and generalizability” section of the revised manuscript.

As the main objective of this study was to test the association between the select SNPs outlined (as proven by the literature to be biologically relevant) and the phenotypes of continued opioid use, relapse, and methadone dose, a focus was placed on examining and reporting these associations within our larger sample population as opposed to replicating the specific findings of previous research, matching their outcome measures and sample population. We have modified the language within our objectives to clarify this intention.

References mentioned in response:

Bawor, M., Dennis, B. B., Tan, C., Pare, G., Varenbut, M., Daiter, J., ... & Samaan, Z. (2015). Contribution of BDNF and DRD2 genetic polymorphisms to continued opioid use in patients receiving methadone treatment for opioid use disorder: an observational study. Addiction science & clinical practice, 10(1), 1-9.

Kamal, F., Flavin, S., Campbell, F., Behan, C., Fagan, J., & Smyth, R. (2007). Factors affecting the outcome of methadone maintenance treatment in opiate dependence. Irish medical journal, 100(3), 393-397.

Hser, Y. I., Saxon, A. J., Huang, D., Hasson, A., Thomas, C., Hillhouse, M., ... & Ling, W. (2014). Treatment retention among patients randomized to buprenorphine/naloxone compared to methadone in a multi‐site trial. Addiction, 109(1), 79-87.

Crist, R. C., Clarke, T. K., & Berrettini, W. H. (2018). Pharmacogenetics of opioid use disorder treatment. CNS drugs, 32(4), 305-320.

How long was each participant on MMT before enrollment in GENOA? The average time on MMT in the GENOA study was 3.7 years (Table 2), but the standard deviation is quite large. Did participants in the current study need to be on MMT for a minimum defined time to be included? If so, what was that minimum time requirement?

Response:

For the GENOA study, there was no minimum duration on MMT required for participants to be included. There was also no minimum amount of time required to be on MMT for the analysis conducted in this current study, so as to not limit the sample size. However, for the outcomes of continued opioid use and relapse, participants had to have a minimum of 3 months’ worth of urine screens while on MMT to be able to quantify these outcomes. 

To account for the variability in the treatment duration across participants, all statistical models adjusted for duration on MMT in months as a covariate. 

The authors state that “primary sources of information used” were “data collected at baseline (enrollment in the study), 3 months prior to study enrollment, and up to 12 months follow up”. So, does the current manuscript encompass only data from a 15 months period that includes 3 months before enrollment into GENOA and 12 months of follow-up of participants after GENOA enrollment? 

Response:

Yes, that is correct. The current manuscript only includes data from urine toxicology screens (UTSs) collected from the GENOA study at 3-month intervals. Since data availability were inconsistent across patients, a duration range of 3 to 15 months was used to prevent the exclusion of patients who did not have more than 3 months’ worth of UTS results recorded. The 15 months period includes data collected 3 months prior to enrollment into GENOA, at baseline, and up to a 12 month follow up after GENOA enrollment, if available. All other data included in this current study were collected at baseline (time of enrollment in the GENOA study). We have revised the language used under the “Study design and setting” section of this manuscript to more clearly explain that. 

Was UTS data available before enrollment of participants in the GENOA study, specifically at 3 months prior to enrollment? 

Response:

These data were available in the medical records for various durations depending on when the participant started the treatment program, however not accessible to the research team until after the participant was recruited and consented for access to medical records. When available, based on if the participants were enrolled in the MMT program for 3 months prior to study enrollment (baseline), UTS data for 3 months prior to enrollment were used to measure outcomes of continued opioid use and relapse for the respective participants.

Were all participants enrolled in GENOA on a stable (maximal) dose of methadone at the time of enrollment, or were some participants enrolled before or during methadone titration up to a stable dose?

Response:

As outlined earlier, not all participants enrolled in the GENOA study were on a stable (maximal) dose at the time of enrollment. Participants were included at any stage of treatment while on MMT, including induction, stabilization, treatment, and tapering. On average, participants have been in the treatment program for 3.7 years. Though this was a limitation due to data unavailability, the duration on MMT and methadone dose variables were accounted for in the statistical models where appropriate. This limitation has been further discussed in the “Limitations and generalizability” section of the revised manuscript.

The authors should indicate how often UTS were done throughout the MMT period. Were they weekly or monthly? 

Response:

UTSs for opioids were performed regularly by clinics on a weekly or sometimes biweekly basis, and then recorded in the GENOA study at 3-month intervals. Please find the UTS frequency information included under the section “Study design and setting” of the revised manuscript.

How were missed UTS appointments handled? Were they treated as missing or as positive? 

Response: 

As we did not have data on missed UTS appointments, any UTS data that were not recorded and represented in the minimum of 3 months amalgamated UTS results in the GENOA study were handled as missing and not included in the analysis. Since data availability were inconsistent across participants, a duration range of 3 to 15 months was used to prevent the exclusion of participants who did not have more than 3 months’ worth of UTS results recorded. 

The definitions of “continued opioid use” and “relapse” do not seem rigorous enough because some level of continued opioid use is anticipated in real world settings of MMT, especially at the start of treatment. The reviewer suggests an analysis using percent positive UTS over time by genotype to determine MMT efficacy (see Crist et al, 2018a) especially at the start of MMT. Similarly, depending on the time on MMT of patients in the current study, the definition of relapse could be defined as more than 1 positive UTS after 3 months of clean UTS because some “breakthrough use” is to be expected as the clinician titrates the methadone dose to an effective level. 

Response: 

The feedback is really appreciated. Though the reviewer’s suggestion is not incorrect, defining continued opioid use as percent positive UTS over time would not be the most ideal way of measuring that variable as continued opioid use is not normally distributed within the MMT patient population, as explained in the comments above. For the reviewers and not included as part of the manuscript, please see below the Q-Q plot and data distribution histogram of continued opioid use measured as a continuous variable of percent positive opioid UTS out of total opioid UTS. As displayed by the histogram, the data are skewed towards low % positive opioid UTS. When tested for normality by the Shapiro-Wilk test and as can be observed by the Q-Q plot, the variable was not normally distributed. The data also could not be transformed into a normal distribution as was evident by log and inverse transformations. Thus, the authors have opted to measure continued opioid use as binary incidence outcomes and analyze the association via a logistic regression.

In terms of the definition of relapse, we have opted to measure incidences of relapse as any positive UTS results following 3 months of negative ones. In doing so, we have opted for a more conservative approach in identifying those who have relapsed. We do acknowledge that there is variability in opioid use trends across patients on MMT and as a result cannot account for all scenarios within our model while accurately representing the population. We have included that as a limitation discussed in our manuscript.

The maximum dose of methadone is a better indication of therapeutic dose than the initial dose of methadone at intake (but, see question above about stable dose of methadone at time of GENOA enrollment). This is especially true for pharmacokinetic metabolic gene analysis, such as CYP2B6.

Response:

We agree that having the maximum dose of methadone would have been a helpful measure in the association with SNPs in the CYP2B6. However, dose at intake for the study does not mean this was the initiation dose. 823 participants have been in the treatment program for 1 year or longer. It is expected that the dose may change during the treatment program, and therefore it is challenging to identify the maximum dose. Though we have tried to account for variability in methadone dose across participants at different stages of MMT by adjusting for participant duration on MMT, we have discussed this limitation under the “Limitations and generalizability” section of the revised manuscript.

Regarding the discussion of SNP rs3745274 (*9 variant of CYP2B6), it is the R-enantiomer of methadone that is the therapeutic enantiomer and binds 50 times more strongly to the MOR (encoded by OPRM1), not the S-enantiomer. Because the authors did not determine the effect of rs3745274 on the R:S-enantiomer ratio, it is speculative to suggest that “[its] effect on continued opioid use in MMT patients could be explained as a decrease in the CYP2B6 gene activity, which could increase plasma methadone concentrations and subside the need for additional opioid intake.” The S-enantiomer is metabolized by CYP2B6, but the *9 variant is predicted to be benign by SIFT and PolyPhen, and was found not to significantly alter mean plasma methadone or the methadone:EDDP ratio (Figure 1 in reference 21 – Ahmad et al, 2017). Also, in Ahmad et al, the significant finding for rs3745274 is likely due to a skewing of the minor allele frequencies in different directions among the controls and methadone groups (overall predicted MAF=27%, controls=22%, methadone=31%) due to sampling error in small group sizes. (The reviewer also points out that the Ahmad et al study considered sex in their statistical analysis). Previous literature (ref 20 – Levran et al, 2013) indicates that it is the CYP2B6*6 haplotype (combined rs2279343 *4 with rs3745274 *9 genotypes) that led to differences in methadone dose requirements. 

Response: 

Thank you for this feedback. We have reworded the language in the discussion section to be more accurate of the findings in the literature and not be speculative. We have also provided a response for the inclusion of sex as a covariate in the literature in a previous comment. 

The authors provide an image of the LD structure for CYP2B6 in their study population, but do not give the actual D’ and r2 values. From the image, it appears that although LD is high between *4 and *9 SNPs, it is not perfect. As such, to attempt to replicate previous findings in the literature, the authors should re-run the statistical analysis using the haplotype of these two SNPs, as was done in Levran et al, 2013.

Response: 

The LD structure for CYP2B6 along with the r-squared values can be found in Figure B of S2 File. The r-squared values are those highlighted within the plot squares. We have revised the figure caption to clarify that. Due to a high r-squared coefficient between rs2279343 (*4) and rs3745274 (*9) of 81, they were deemed to be in high LD and were treated as such. Though replicating findings of CYP2B6 haplotype analyses would have been interesting, conducting a haplotype analysis for this study would not provide additional information on variant-phenotype associations as SNPs rs2279343 and rs3745274 (which was tested) are in high LD.

What statistics program did the authors use to impute the missing covariate data? How many imputations did the authors run? If missing data imputations were run in R, the authors should state the package they used – for instance, did they use ‘norm’? Could the authors have instead used the 'keep-pheno-on-missing-cov' option for covariates in PLINK 1.07? 

Response:

No statistical package was used for the missing phenotypic data imputations. Averages of the duration on MMT and methadone dose values were used to substitute for any missing values in excel. The term ‘mean substitution’ has been used to clarify the meaning behind imputed variables within this manuscript. 

Did the authors test models other than the ‘Genotypic’ (ADD) model? Other genetic models were used in previous literature (see Crist et al, 2018a). 

Response:

Within the scope of this manuscript, only an additive model was used to test statistical associations. As opioid use and MMT response are complex traits with multiple genes and multiple variants of a gene contributing to their outcomes, an additive model was seen as best fitting.

The referenced Crist et al study has also reported using an additive genetic model to test the association between the rs10485058 genotype with their outcome of self-reported relapse. 

For covariates that were not significant in the full model, the authors should re-run the statistics without them in the model to determine whether the significance of the outcome measures change.

Response: 

Thank you for the suggestion, however, we retained these covariates for their clinical relevance and not statistical significance. If we are to exclude these variables, the results may be confounded. 

Minor issues:

Some of the references are incorrect. For example, reference 32 in the manuscript is supposed to reference R (page 11, line 208) but is listed in the references as Cahn et al, 2011, J. Wildl Manage 75(8):1753-66. Was this supposed to be reference 33 for R as listed in the references section?

Response: 

Thank you for highlighting this. The references list and in-text citations have been fixed to match the references. Please note that due to this change being implemented through an imbedded reference manager, the change in reference numbers will not be viewed as tracked. 

The authors should state which analyzed SNPs were directly genotyped (on the chip) and which were imputed.

Response:

We have included data on which SNPs from those of interest were imputed in the S2 File. 

SNPs that were directly genotyped include: rs10485058, rs1799971, rs8192719, rs3745274.

SNPs that were imputed include: rs73568641, rs7451325, rs2279343, rs10403955.

The authors should clarify what the reference condition was for each statistical analysis. For instance, was the reference condition defined as an individual ‘having the minor allele and being positive for the particular outcome measure’, or was it ‘having the minor allele and not being positive for the outcome measure’?

Response:

All results reported in this manuscript, as outlined in Table 3 and Table 4, test for the minor allele. The table legends specify that the minor allele is the reference and tested allele. In terms of outcomes, the statistical analyses test for having the minor allele and being positive for a phenotype (i.e. cases) in the logistic regressions. The language under the “Statistical methods” section has been revised to reflect this. 

References mentioned (but not referenced in the manuscript):

Crist RC, Doyle GA, Nelson EC, Degenhardt L, Martin NG, Montgomery GW, Saxon AJ, Ling W, Berrettini WH. A polymorphism in the OPRM1 3'-untranslated region is associated with methadone efficacy in treating opioid dependence. Pharmacogenomics J. 2018a Jan;18(1):173-179. doi: 10.1038/tpj.2016.89.

Crist RC, Li J, Doyle GA, Gilbert A, Dechairo BM, Berrettini WH. Pharmacogenetic analysis of opioid dependence treatment dose and dropout rate. Am J Drug Alcohol Abuse. 2018b ;44(4):431-440. doi: 10.1080/00952990.2017.1420795.

Reviewer #3: This paper aims to test associations between several OPRM1 and CYP2B6 SNPs on several methadone maintenance treatment outcomes in males and females separately.

The strength of the study is the large sample size as well as the high number of females, which is rare in MMT studies. Contradictory findings have been published on these SNPs, mostly in smaller studies, thus a large replicate study is interesting.

My main concerns of the manuscript are as follows:

- In the abstract and in the results sections (also in the tables), the results are always described in such a way that they first seem to be statistically significant, while they are not.

Response:

Thank you for highlighting this. We have revised the language in the abstract and results sections to be more reflective of the actual results of the study. 

- Most included patients were taking methadone but it seems from table 2 that a few took suboxone. Why they were not excluded as the study is on MMT?

Response:

Table 2 showcases the overall study sample that was collected as part of the GENOA study and had genotyped data available (please refer to Figure A in S2 File for a more detailed flow diagram of the sample size). Participants included in this study’s analyses, however, were only those on methadone maintenance treatment (n=1172) as the sample size of those on suboxone was too small for analysis (n=52). Anyone who was administered suboxone was excluded, as mentioned under the section “Eligibility criteria”. For the purpose of showcasing the data availability for this sample, we have displayed numbers of participants from the GENOA study who were administered methadone versus suboxone in Table 2. We have corrected the abstract to clarify that 1172 participants treated with methadone were included in this study. 

 - For the outcomes on continued opioid use and relapse, they should be clarified. The frequency of UTS should be indicated to better describe these outcomes. Is the duration of 3 to 15 months for all patients?

Response: 

Urine toxicology screens (UTSs) for opioids were performed regularly on a weekly or sometimes biweekly basis, and then recorded in the GENOA study at 3-month intervals. Since data availability were inconsistent across patients, a duration range of 3 to 15 months was used to prevent the exclusion of patients who did not have more than 3 months’ worth of UTS results recorded. The mean total number of opioid UTSs is 74.34 (+/- 34.64), as reported in Table 2. Incomplete data could have been a result of patient incarcerations, relocations, hospitalizations, mortality, or other unaccounted outcomes.

To best account for the differences in the duration of opioid UTSs, duration on MMT was included as a covariate in all analysis models.

Please find the UTS frequency information also included under the section “Study design and setting”. 

- For the methadone dose outcome, more information on how long the dose was unchanged before inclusion should be added. Detailed descriptive statistics of the dose should be added (range, normal distribution in the population, …).

Response:

Though reporting on how long the dose was unchanged prior to participant inclusion in the study would have been informative, that information is unfortunately unavailable. At the time of enrollment in the study, participants were asked if their dose had remained stable for the past 3 months or if it had changed. Because the yes/no responses were self-reported, we cannot ascertain that they were accurate or be sure of the direction of that change (increased or decreased dose). As a result, this measure was not found to be informative and was not included in the manuscript. Also, methadone doses have the potential to change on a weekly basis, as clinically indicated. We acknowledge that the stability of the methadone dose is a limitation of the study and have included that in the limitations’ discussion of the manuscript. 

We have also revised Table 2 to include more detailed descriptive statistics on the methadone dose, including the range of methadone doses reported.

- The outcome of the methadone dose is questionable as methadone metabolism displays a high interindividual variability and as patients might be in different phase of their MMT (treatment introduction, stabilization or slow tapering of the dose for example). Also, in certain MMT prescribing center, there is a maximal dose not be exceeded but that could remain insufficient for certain patients who therefore continue to use opiates

Response:

That is a great point. We agree that there is interindividual variability in the level of methadone metabolism across patients and did anticipate that to be represented in the association between the CYP2B6 gene SNPs and the methadone dose outcome. 

With regards to patients being in different phases of their MMT, we unfortunately were not able to precisely account for that, although on average patients were in treatment for 3.7 years, and have thus reported it as a limitation of the study. Though as mentioned earlier, we did account for the patients’ duration on MMT in the statistical models.

Regarding the maximal dose of methadone provided by treatment centers, we do acknowledge that there are certain maximal doses that cannot be exceeded and that the methadone dose affects treatment outcomes of patients, and have as such accounted for methadone dose as a covariate in the measure of continued opioid use. 

In general, when discussing the individual SNPs, it will help to include the gene name before the rs number (ex. OPRM1 rs73568641)

Response:

We have revised the manuscript text where appropriate to include the gene name before outlining the rs-ID of the SNP of interest.

In the study design and setting section (p.5, line 107): it is not clear which data was collected 3 months prior to study enrollment.

Response:

Only UTS results were collected 3 months prior to study enrollment. We have revised the language under the “Study design and setting section” to clarify that. 

In Table 2, the significantly different variables between male and female should be indicated (*). The opioid prescription variable should be clarified.

Response:

Differences between the male and female variables shown in Table 2 were calculated, p-values displayed, and significant differences outlined with an asterisk. We have further revised the “opioid prescription” variable to say “Participant taking opioid prescription” for clarity.

In the discussion (p.16, line 258): no trend was observed for rs73568641 on relapse in female.

Response:

Thank you for highlighting that. We have revised the text in the discussion to reflect the results of the study more accurately.

---

## [Decision Letter · Decision Letter 1]

12 Oct 2021

PONE-D-21-11145R1Implications of *OPRM1* and *CYP2B6* variants on treatment outcomes in methadone-maintained patients in Ontario: Exploring sex differencesPLOS ONE

Dear Dr. Samaan,

Thank you for submitting your manuscript to PLOS ONE. After careful consideration, we feel that it has merit but does not fully meet PLOS ONE’s publication criteria as it currently stands. Therefore, we invite you to submit a revised version of the manuscript that addresses the points raised during the review process.

Please address a couple of very minor issues from Reviewer 3.

We look forward to receiving your revised manuscript.

Kind regards,

Huiping Zhang

Academic Editor

PLOS ONE

Journal Requirements:

Reviewers' comments:

Reviewer's Responses to Questions

**Comments to the Author**

1. If the authors have adequately addressed your comments raised in a previous round of review and you feel that this manuscript is now acceptable for publication, you may indicate that here to bypass the “Comments to the Author” section, enter your conflict of interest statement in the “Confidential to Editor” section, and submit your "Accept" recommendation.

Reviewer #1: All comments have been addressed

Reviewer #3: (No Response)

2. Is the manuscript technically sound, and do the data support the conclusions?

Reviewer #1: Partly

Reviewer #3: Yes

3. Has the statistical analysis been performed appropriately and rigorously? 

Reviewer #1: Yes

Reviewer #3: Yes

4. Have the authors made all data underlying the findings in their manuscript fully available?

Reviewer #1: Yes

Reviewer #3: Yes

5. Is the manuscript presented in an intelligible fashion and written in standard English?

Reviewer #1: Yes

Reviewer #3: Yes

6. Review Comments to the Author

Reviewer #1: (No Response)

Reviewer #3: Thank you for your response.

My comments have been addressed except for the addition of the range (min-max or interquartile) of methadone dose in Table 2. The number of significant digits (for methadone dose but not only) used in table 2 should also be reviewed.

And there is an inversion in table 2 in the Continued opioid use frequency.

7. PLOS authors have the option to publish the peer review history of their article (what does this mean?). If published, this will include your full peer review and any attached files.

Reviewer #1: **Yes: **Wade Berrettini

Reviewer #3: No

---

## [Author Response · Author response to Decision Letter 1]

25 Nov 2021

Reviewer #1: (No Response)

Reviewer #3: Thank you for your response.

My comments have been addressed except for the addition of the range (min-max or interquartile) of methadone dose in Table 2. The number of significant digits (for methadone dose but not only) used in table 2 should also be reviewed.

And there is an inversion in table 2 in the Continued opioid use frequency.

Response: Thank you for your comments and review. In response to Reviewer #3’s comment, the range that was included in the previous revision as the difference between the highest and lowest methadone doses in Table 2 has been modified to show the “minimum-maximum” values. Table 2 values (of methadone dose and other descriptors) have been edited where applicable to show significant digits that are proportional to the magnitude of the SD values presented. Finally, the inversion in the “Continued opioid use” frequency in Table 2 has been corrected; thank you for noting that.

---

## [Editor Report · Decision Letter 2]

29 Nov 2021

Implications of *OPRM1* and *CYP2B6* variants on treatment outcomes in methadone-maintained patients in Ontario: Exploring sex differences

PONE-D-21-11145R2

Dear Dr. Samaan,

We’re pleased to inform you that your manuscript has been judged scientifically suitable for publication and will be formally accepted for publication once it meets all outstanding technical requirements.

Kind regards,

Huiping Zhang

Academic Editor

PLOS ONE
---

## [Editor Report · Acceptance letter]

6 Dec 2021

PONE-D-21-11145R2 

Implications of *OPRM1* and *CYP2B6* variants on treatment outcomes in methadone-maintained patients in Ontario: Exploring sex differences 

Dear Dr. Samaan:

I'm pleased to inform you that your manuscript has been deemed suitable for publication in PLOS ONE. Congratulations! Your manuscript is now with our production department. 

Kind regards, 

on behalf of

Dr. Huiping Zhang 

Academic Editor

PLOS ONE